# Intracellular complexities of acquiring a new enzymatic function revealed by mass-randomisation of active-site residues

Kelsi R Hall[1,2], Katherine J Robins[1†], Elsie M Williams[1,2], Michelle H Rich[1‡], Mark J Calcott[1,2], Janine N Copp[1§], Rory F Little[1#], Ralf Schwörer[2,3], Gary B Evans[2,3], Wayne M Patrick[1,2], David F Ackerley[1,2]*

[1]School of Biological Sciences, Victoria University of Wellington, Wellington, Wellington, New Zealand; [2]Centre for Biodiscovery, Victoria University of Wellington, Wellington, New Zealand; [3]Ferrier Institute, Victoria University of Wellington, Wellington, New Zealand

*For correspondence: david.ackerley@vuw.ac.nz

Present address: †The Manchester Institute of Biotechnology, The University of Manchester, Manchester, United Kingdom; ‡BiOrbic, SFI Bioeconomy Research Centre, University College Dublin, Belfield, Ireland; §Michael Smith Laboratories, University of British Columbia, Vancouver, Canada; #Leibniz Institute for Natural Product Research and Infection Biology, Hans Knöll Institute, Jena, Germany

Competing interests: The authors declare that no competing interests exist.

**Abstract** Selection for a promiscuous enzyme activity provides substantial opportunity for competition between endogenous and newly-encountered substrates to influence the evolutionary trajectory, an aspect that is often overlooked in laboratory directed evolution studies. We selected the *Escherichia coli* nitro/quinone reductase NfsA for chloramphenicol detoxification by simultaneously randomising eight active-site residues and interrogating ~250,000,000 reconfigured variants. Analysis of every possible intermediate of the two best chloramphenicol reductases revealed complex epistatic interactions. In both cases, improved chloramphenicol detoxification was only observed after an R225 substitution that largely eliminated activity with endogenous quinones. Error-prone PCR mutagenesis reinforced the importance of R225 substitutions, found in 100% of selected variants. This strong activity trade-off demonstrates that endogenous cellular metabolites hold considerable potential to shape evolutionary outcomes. Unselected prodrug-converting activities were mostly unaffected, emphasising the importance of negative selection to effect enzyme specialisation, and offering an application for the evolved genes as dual-purpose selectable/counter-selectable markers.

## Introduction

Many (if not all) enzymes are promiscuous, meaning that in addition to their primary biological role(s) they can catalyse minor side reactions that have no apparent physiological relevance, either because they are too inefficient or because the substrate is not naturally encountered (*Copley, 2015*). From an evolutionary perspective, promiscuity can play an important role in contingency, providing a reservoir of potential functions that a cell can tap in response to changing circumstances (*O'Brien and Herschlag, 1999*; *Copley, 2015*). As demonstrated by the emergence of resistance to xenobiotic pollutants or clinical antibiotics (*O'Brien and Herschlag, 1999*; *Hall, 2004*; *Ramos et al., 2005*; *Copley, 2009*; *Khersonsky and Tawfik, 2010*), a strong selection pressure can cause latent promiscuous activities to be rapidly amplified to physiologically relevant levels (*Newton et al., 2015*).

Catalytic transitions to an alternate substrate have been modelled experimentally using iterative rounds of random mutagenesis (e.g. error-prone PCR (epPCR)), a powerful directed evolution strategy that enables adaptive landscapes to be explored under defined laboratory conditions (*Kaltenbach et al., 2015*; *Kaltenbach et al., 2016*). These laboratory evolution studies have indicated that selection for substantial increases in a promiscuous activity typically results in only weak trade-offs against the native activity; and therefore, the transition from one primary function to

**eLife digest** In the cell, most tasks are performed by big molecules called proteins, which behave like molecular machines. Although proteins are often described as having one job each, this is not always true, and many proteins can perform different roles. Enzymes are a type of protein that facilitate chemical reactions. They are often specialised to one reaction, but they can also accelerate other side-reactions. During evolution, these side-reactions can become more useful and, as a result, the role of the enzyme may change over time.

The main role of the enzyme called NfsA in *Escherichia coli* bacteria is thought to be to convert molecules called quinones into hydroquinones, which can protect the cell from toxic molecules produced in oxidation reactions. As a side-reaction, NfsA has the potential to protect bacteria from an antibiotic called chloramphenicol, but it generally does this with such low efficacy that the effects are negligible. Producing hydroquinones is helpful to the cell in some situations, but if bacteria are regularly exposed to chloramphenicol, NfsA's role aiding antibiotic resistance could become more important. Over time, the enzyme could evolve to become better at neutralising chloramphenicol. Therefore, NfsA provides an opportunity to study the evolution of proteins and how bacteria adapt to antibiotics.

To see how evolution might affect the activity of NfsA, Hall et al. generated 250 million *E. coli* with either random or targeted changes to the gene that codes for the NfsA enzyme. The resulting variants of NfsA that were most effective against chloramphenicol all had a change that eliminated the enzyme's ability to convert quinones. This result demonstrates a key trade-off between roles for NfsA, where one must be lost for the other to improve.

These results demonstrate the interplay between a protein's different roles and provide insight into bacterial drug resistance. Additionally, the experiments showed that the bacteria with improved resistance to chloramphenicol also became more sensitive to another antibiotic, metronidazole. These findings could inform the fight against drug-resistant bacterial infections and may also be helpful in guiding the design of proteins with different roles.

another tends to progress via generalist enzyme intermediates (*Kaltenbach et al., 2016*). Two leading teams have offered contrasting hypotheses to explain this phenomenon. In 2005, Tawfik and co-workers proposed that enzymes possess an innate 'robustness' and stability that buffers them against the potentially detrimental effects of novel mutations, coupled with a 'plasticity' that can amplify promiscuous functions with relatively few mutations (*Aharoni et al., 2005*). More recently, Tokuriki and co-workers demonstrated that a robust native activity is not a prerequisite for weak trade-offs, and suggested that the predominance of these in the literature may instead be artefactual; a consequence of laboratory evolution studies being highly biased towards strong selection for a new function without any selection against the native activity (*Kaltenbach et al., 2016*). They argue that it is unclear how specialisation can occur in this manner, and that in nature, selection might frequently exist to erode the original function. By way of example, they offer a scenario where the native and new substrate compete for the same active site (*Kaltenbach et al., 2016*).

In addition to their exclusive emphasis on positive selection, we note that the studies overviewed by the Tokuriki team were also heavily biased towards heterologous enzyme expression and/or a transition in activity from one exogenously applied substrate to another (*Kaltenbach et al., 2016*). Thus, there has been little consideration of how the native substrate might influence the evolutionary trajectory. We were therefore motivated to study the selection of a promiscuous function within the native host environment, with particular focus on the key catalytic changes driving the transition. Recognising that the stochastic nature of iterative random mutagenesis is unlikely to yield the most efficient pathway to a selected outcome, we sought to implement simultaneous mass-mutagenesis on a massive scale that would allow us to retrospectively assess all possible intermediates of our top variants, and infer the most plausible stepwise evolutionary trajectories. We were able to achieve both these goals by employing the *Escherichia coli* nitro/quinone reductase NfsA as a new model system that offers several key advantages. NfsA is a member of a large bacterial superfamily comprising highly promiscuous FMN-dependent oxidoreductases that accept electrons from NAD(P)H and transfer them to a diverse range of substrates (*Williams et al., 2015*; *Akiva et al., 2017*).

Expression of *nfsA* is governed by the *soxRS* regulon, and NfsA is thought to guard against oxidative stress through reduction of water-soluble quinones such as 1,4-benzoquinone (*Liochev et al., 1999*; *Paterson et al., 2002*; *Copp et al., 2017*). Although most efficient with quinone substrates, NfsA is also able to reduce a wide diversity of nitroaromatic compounds (*Valiauga et al., 2017*). This is generally believed to represent non-physiological substrate ambiguity, as there are relatively few nitroaromatic natural products, and in many cases nitro-reduction yields a more toxic derivative (*Winkler and Hertweck, 2007*; *Parry et al., 2011*; *Williams et al., 2015*). An important exception is that nitro-reduction of chloramphenicol transforms this antibiotic to a product that is not discernibly toxic to bacteria (*Yunis, 1988*; *Erwin et al., 2007*; *Crofts et al., 2019*). We have observed that over-expressed native NfsA confers only slight chloramphenicol protection to *E. coli* host cells, but reasoned that we could select for improved detoxification in an extremely high-throughput manner by plating variant libraries on chloramphenicol-amended media. Because members of the bacterial nitroreductase superfamily appear to have unusually plastic active sites (*Akiva et al., 2017*), we considered that simultaneous mass-mutagenesis of up to eight active-site residues should be possible. In effect, we aimed to strip NfsA of its engine, and then select for a superior configuration of parts assembled within the empty chassis. By leaping directly to a new fitness peak, we considered that we might arrive at synergistic combinations of substitutions that would be difficult to achieve by iterative random mutagenesis approaches.

## Results

### Design of an eight randomised codon *nfsA* gene library

We have previously conducted several different mutagenesis studies on *nfsA*, seeking to enhance activity with prodrugs and/or positron emission tomography (PET) imaging probes for cancer gene therapy applications (*Williams, 2013*; *Copp et al., 2017*; *Rich, 2017*), or to assess potential collateral sensitivities between niclosamide and the antibiotics metronidazole and nitrofurantoin (*Copp et al., 2020*). Based on this previous work we empirically identified eight active-site residues (S41, L43, H215, T219, K222, S224, R225, and F227; *Figure 1A*) as being individually mutable and having the potential to contribute to generically improved nitroreductase activity. We then designed a degenerate gene library to enable simultaneous randomisation of each residue. As complete randomisation of target codons (e.g. NNK degeneracy) would have yielded an impractically large library of >$10^{12}$ ($32^8$ or more) gene variants, we instead used a restricted set (*Figure 1B*). The degenerate codon NDT was preferred at most sites, as this specifies 12 different amino acids that represent a balanced portfolio of small and large, polar and non-polar, aliphatic and aromatic, and negatively and positively charged side chains (*Reetz et al., 2008*). However, at positions 219 and 222, NDT codons did not include the native NfsA residue as an option, so the alternative degeneracies NHT (12 unique amino acids) and VNG (11 unique amino acids) were chosen as acceptably balanced alternatives (*Figure 1B*). In total, our library represented 430 million possible gene combinations, collectively specifying 394 million different NfsA variants.

### Selection and characterisation of superior chloramphenicol-detoxifying NfsA variants

Following artificial synthesis and cloning, our library was used to transform *E. coli* 7NT cells (a strain in which endogenous nitroreductase genes had been deleted). We ultimately recovered a total of 398 million transformed colonies, a collection predicted by GLUE (*Firth and Patrick, 2008*) to represent 252 million different NfsA variants. Despite the drastic reconfiguration of their encoded active sites, a surprising 0.05% of the gene variants (~200,000 clones) were more effective than wild-type *nfsA* (i.e. enabled colony formation on LB agar amended with 3 µM chloramphenicol, the lowest concentration at which wild-type *nfsA* was unable to support host cell growth). This robust tolerance to active-site randomisation confirmed that NfsA exhibits a substantial degree of active-site plasticity.

We next plated the library on ≥45 µM chloramphenicol, recovering a total of 365 colonies. Retransformation of fresh 7NT host cells with the variant-encoding plasmids was performed to eliminate any selected chromosomal mutations, followed by validation of activities in liquid growth assays. Sequencing and elimination of duplicates yielded 30 top variants that exhibited evidence of a conserved genetic response to the chloramphenicol selection. Particularly strong trends were

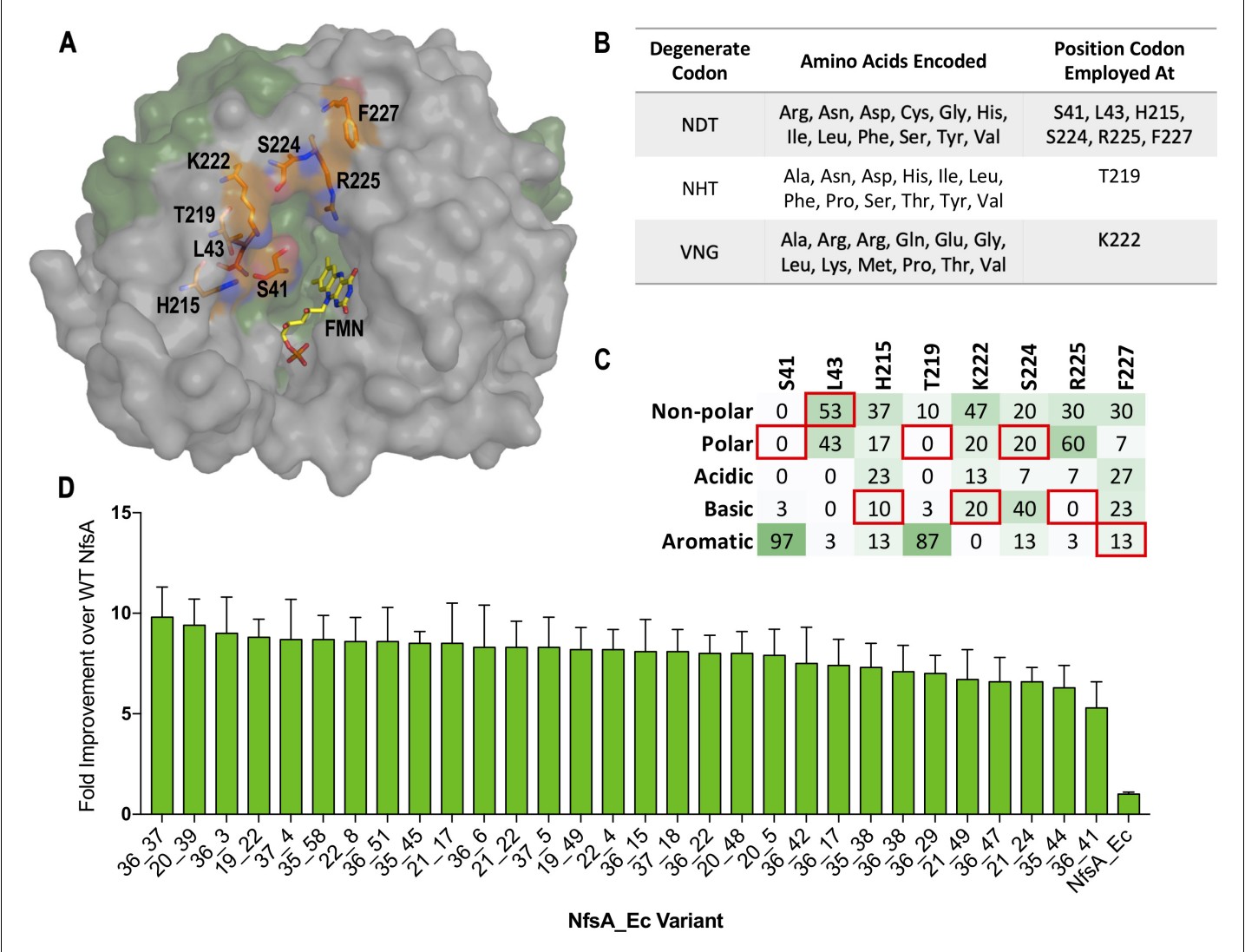

**Figure 1.** Creation, selection, and characterisation of 30 top chloramphenicol-detoxifying NfsA variants. (A) Structure of NfsA, based on PDB 1f5v. One monomer is shown in grey and one monomer in green. The eight residues simultaneously targeted in NfsA (carbons highlighted in orange) and the FMN cofactor (carbons highlighted in yellow) are shown in stick form. For clarity, only one of the two FMN-binding active sites in the enzyme homodimer is portrayed. (B) Summary of the amino acid repertoire encoded by each degenerate codon. (C) Percentage of the five amino acid side-chain categories at each of the eight targeted positions for the top 30 chloramphenicol reducing variants (a complete summary of all residue substitutions is provided in *Supplementary file 1a*). The property of the native amino acid at each position is boxed in red. (D) Fold improvement in chloramphenicol EC$_{50}$ values for *E. coli* 7NT strains expressing the top 30 chloramphenicol-detoxifying *nfsA* variants over the native *nfsA* control (far right). Data presented in D represents the average of at least four biological repeats ± 1 S.D.

The online version of this article includes the following source data and figure supplement(s) for figure 1:

**Source data 1.** Source data for *Figure 1*.
**Figure supplement 1.** Relative enzyme expression levels for the top 30 chloramphenicol-detoxifying NfsA variants generated by multi-site saturation mutagenesis.

observed at positions 41 and 219 (where the native serine or threonine was substituted by an aromatic residue in ≥26 of the 30 cases), and at position 225 (100% substitution of the native arginine by an uncharged or acidic residue) (*Figure 1C*, *Supplementary file 1a*). Only at position 43 was the native leucine or a chemically similar residue frequently retained (16/30 cases). In EC$_{50}$ assays, 7NT cells expressing these 30 variants demonstrated nearly 6- to 10-fold greater chloramphenicol tolerance than those expressing native *nfsA* (*Figure 1D*). SDS-PAGE analysis revealed that there was no substantial variation in expression between these variants and native *nfsA*, and hence the enhanced

chloramphenicol detoxification was not a consequence of elevated expression levels (*Figure 1—figure supplement 1*).

To evaluate the impact of the active-site reconfiguration on catalytic activity, the top five chloramphenicol-detoxifying NfsA variants were purified as His$_6$-tagged proteins and evaluated in steady-state kinetics assays. We were surprised to discover that these variants exhibited only marginal (at most 2.2-fold) improvements in chloramphenicol $k_{cat}/K_M$ over wild-type NfsA (*Table 1*). However, in every case the variants were impaired in $k_{cat}$ (6–10-fold lower than NfsA) but greatly improved in $K_M$ (8–13-fold lower than NfsA). Thus, it appeared that the in vivo improvements in chloramphenicol detoxification were driven primarily by enhanced substrate affinity.

We previously observed a similar phenomenon when using an epPCR strategy to evolve NfsA for improved activation of the anti-cancer prodrug PR-104A, with all top variants exhibiting a lower $k_{cat}$ and lower $K_M$ for PR-104A, and none being significantly improved in $k_{cat}/K_M$ over the native enzyme (*Copp et al., 2017*). In that study, we postulated that the improved in vivo activities were a consequence of diminished competitive inhibition by native quinone substrates present in the *E. coli* cytoplasm; although the top variant was still active with 1,4-benzoquinone, we found its PR-104A reduction activity was less affected by addition of 1,4-benzoquinone to the reaction mix than was the case for wild-type NfsA (*Copp et al., 2017*). Here, we were unable to perform the same in vitro competition assays, as both 1,4-benzoquinone and chloramphenicol reduction are monitored by following NADPH depletion at 340 nm. However, when assayed individually we found that 1,4-benzoquinone reduction was unmeasurable in each of the five top chloramphenicol-detoxifying variants (*Table 1*). In contrast, for wild-type NfsA, 1,4-benzoquinone reduction ($k_{cat}/K_M = 5.8 \times 10^6$; *Valiauga et al., 2017*) is nearly 1000-fold more efficient than chloramphenicol reduction (*Table 1*). Our data are therefore consistent with in vivo chloramphenicol reduction having been amplified for the selected variants by the elimination of competitive quinone inhibition.

## Recreating all hypothetical evolutionary trajectories for the top chloramphenicol-detoxifying NfsA variants

We next sought to probe the contributions to improved chloramphenicol detoxification and/or diminished 1,4-benzoquinone reduction made by key substitutions, or combinations thereof. The top two chloramphenicol-detoxifying variants (on the basis of EC$_{50}$ measurements; 36_37 and 20_39) each had seven substitutions at the eight targeted positions, with both containing the wild-type residue leucine at position 43 (36_37 = S41Y, H215C, T219Y, K222V, S224R, R225V, F227G; 20_39 = S41Y, H215N, T219Y, K222R, S224Y, R225D, F227H; *Figure 2A–C*). Therefore, there are 126 possible intermediate forms ($2^7 – 2$) between wild-type NfsA and each selected variant. Genes encoding the 126 intermediate forms for each variant were artificially synthesised, cloned and

**Table 1.** Kinetic parameters of chloramphenicol reduction and turnover rate of 1,4-benzoquinone for purified NfsA variants (the top five by in vivo EC$_{50}$ ranking).

Apparent $K_M$ and $k_{cat}$ at 250 µM NADPH were calculated using Graphpad 8.0. Kinetic parameters could not be accurately determined for 1,4-benzoquinone for any of the selected variants; in an attempt to detect trace activities, the catalytic rate of 1,4-benzoquinone reduction was measured at a single high concentration of 1,4-benzoquinone (100 µM), with reactions initiated by addition of 250 µM NADPH. All reactions were measured in triplicate and errors are ±1 S.D. *Apparent $k_{cat}$ and $K_M$ as determined at 250 µM NADPH. **Measured rates following addition of 250 µM NADPH. ***N.D.=not detectable (<0.1 s$^{-1}$).

| Variant | $k_{cat}$ (s$^{-1}$)* | $K_M$ (µM)* | $k_{cat}/K_M$ (M$^{-1}$·s$^{-1}$) | Fold Improvement over NfsA | Turnover rate of 100 1,4-benzoquinone (s$^{-1}$)** |
|---|---|---|---|---|---|
| NfsA | 0.89 ± 0.03 | 1000 ± 100 | 860 ± 90 | 1.0 | 9.0 ± 0.5 |
| 36_37 | 0.14 ± 0.004 | 130 ± 20 | 1100 ± 200 | 1.3 | N.D.*** |
| 20_39 | 0.16 ± 0.004 | 80 ± 10 | 1900 ± 200 | 2.2 | N.D.*** |
| 36_3 | 0.11 ± 0.003 | 130 ± 10 | 850 ± 90 | 1.0 | N.D.*** |
| 37_4 | 0.09 ± 0.004 | 80 ± 20 | 1000 ± 200 | 1.2 | N.D.*** |
| 19_22 | 0.13 ± 0.003 | 90 ± 10 | 1500 ± 200 | 1.7 | N.D.*** |

The online version of this article includes the following source data for Table 1:
Source data 1. Source data for *Table 1*.

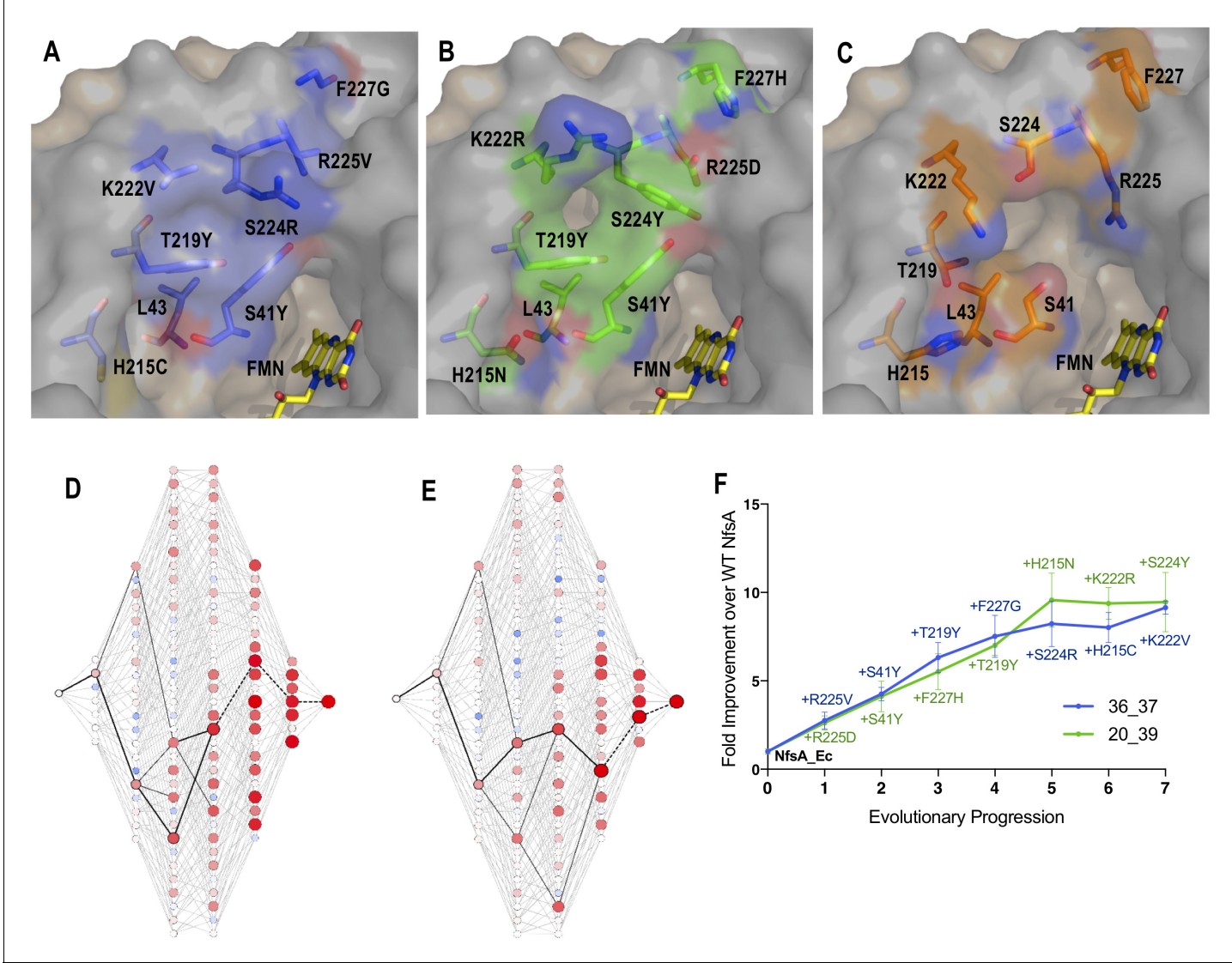

**Figure 2.** Recreating the hypothetical evolutionary trajectories of NfsA variants 36_37 and 20_39. (A–C) Residues in the active site of 36_37 (A, blue), 20_39 (B, green) and wild-type NfsA (C, orange); based on PDB 1f5v (*Kobori et al., 2001*). In each panel, one NfsA monomer is shaded grey and the other is tan. The orientation of the mutated residues in 36_37 and 20_39 was predicted using the mutagenesis wizard on PyMOL, which selected the most likely rotamer conformation based on the frequencies of occurrence in proteins while avoiding clashes with other residues. (D–E) All 5040 hypothetical evolutionary trajectories of 36_37 (D) or 20_39 (E). Black lines represent primary paths in which each step resulted in a a > 16% increase in chloramphenicol detoxification. Thick black lines represent the most probable stepwise evolutionary trajectory as explained in Figure F. The colour and diameter of nodes corresponds to the fold improvement in chloramphenicol detoxification over wild-type NfsA (blue/smaller = less active, red/larger = more active). A larger version of each image is provided in *Figure 2—figure supplement 1* and *Figure 2—figure supplement 2*. (F) The most plausible stepwise evolutionary trajectory for each of variant 36_37 (blue) and variant 20_39 (green). To establish these, the substitution which resulted in the greatest improvement in chloramphenicol detoxification over wild-type NfsA (WT NfsA) was selected at each point in the hypothetical evolutionary progression. If no substitutions improved chloramphenicol detoxification, then the substitution was selected which resulted in the smallest decrease in activity (shown as a dotted black line in D-E). Data presented in (D-F) represent the average of at least four biological repeats ± 1 S.D. The online version of this article includes the following source data and figure supplement(s) for figure 2:

**Source data 1.** Source data for *Figure 2*.
**Figure supplement 1.** Evolutionary network of 36_37.
**Figure supplement 2.** Evolutionary network of 20_39.

expressed in *E. coli* 7NT cells, and the chloramphenicol tolerance of the resulting strains assessed in $EC_{50}$ growth assays. A Python script was generated to delineate all 5040 (7!) possible evolutionary trajectories and the output used to generate full network graphs (*Figure 2D–E*, *Figure 2—figure supplement 1*, *Figure 2—figure supplement 2*). We then considered whether traditional stepwise directed evolution strategies, which require each substitution to directly improve the selected activity (e.g. across iterative rounds of epPCR), could have plausibly generated either of variants 36_37 or 20_39. For the purposes of this analysis we considered 'improvement' to be a > 16% increase in chloramphenicol $EC_{50}$ for each step, as this was the average error across all $EC_{50}$ measurements. In neither case was there a clear path from NfsA to the final variant that involved exclusively upward steps in the hypothetical evolutionary trajectory (*Figure 2D–F*). Nevertheless, it was evident that the final two substitutions (H215C and K222V for variant 36_37, and K222R and S224Y for variant 20_39) did not contribute substantially to the overall chloramphenicol-detoxifying activity of each variant. Thus, we concluded that iterative evolutionary strategies could have plausibly generated NfsA variants exhibiting similar levels of chloramphenicol-detoxifying activity to 36_37 and 20_39, but that there were very few accessible pathways for this (*Figure 2D–E*).

The dearth of accessible hypothetical evolutionary pathways suggested extensive epistasis, a phenomenon that several teams have previously observed when evolving enzymes (*Weinreich et al., 2005*; *Poelwijk et al., 2011*; *Kaltenbach, 2014*; *Yang et al., 2019*; *Ben-David et al., 2020*), where the fitness effects of certain substitutions only manifest when other substitutions have already been made. Most prominently, we noted that only one of the seven substitutions present in each of variants 36_37 and 20_39 significantly enhanced chloramphenicol detoxification when introduced on an individual basis (*Figure 3A*). Although it was the same residue, R225, that was substituted in each case, the substituting residues possessed very different chemical properties (negatively charged aspartate in 20_39 *versus* non-polar valine in 36_37). This, together with the observation that none of our top 30 selected variants had retained an arginine at position 225 (*Figure 1C*, *Supplementary file 1a*), suggested that it was important for arginine 225 to be eliminated before the other active-site substitutions could make a discernible contribution to improved chloramphenicol detoxification.

We also found evidence of higher-order epistasis beyond the requirement for elimination of R225. For example, both of the top selected variants contained the substitutions S41Y and T219Y,

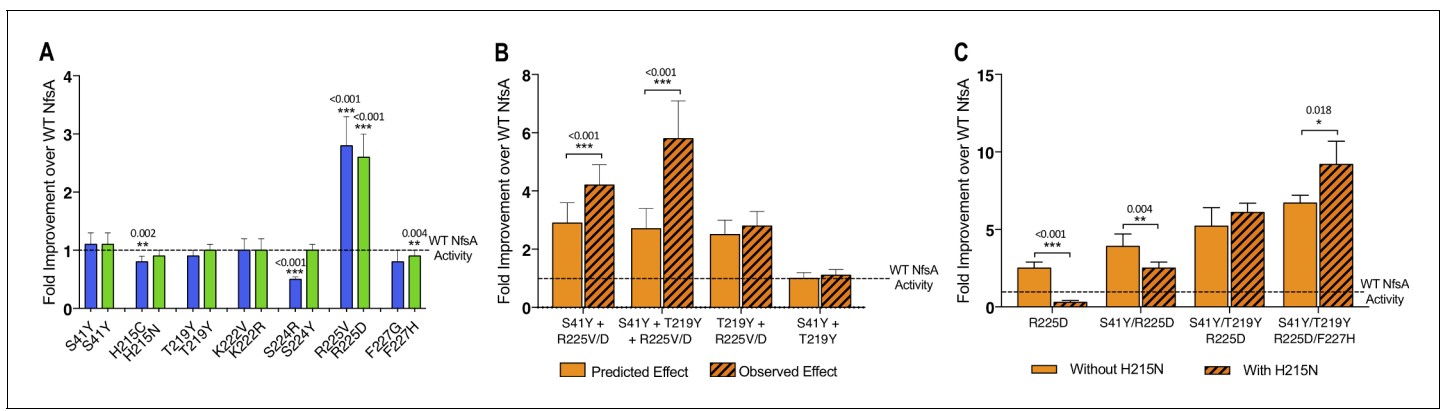

**Figure 3.** Complex epistatic interactions exist in 36-37 and 20_39. (**A**) The effect on chloramphenicol detoxification of introducing individual substitutions present in 36_37 (blue) and 20_39 (green) into wild-type (WT) NfsA. (**B**) Observation of epistatic interactions between S41Y, T219Y and R225V/D. The predicted multiplicative effects (solid bars) were calculated by multiplying the fold-increase conferred by individual amino acid substitutions. The error of the predicted effects was derived using an error propagation equation ($\delta R = R \times \sqrt{\left(\frac{\delta X}{X}\right)^2 + \left(\frac{\delta Y}{Y}\right)^2 + \left(\frac{\delta Z}{Z}\right)^2}$ where $\delta X$, $\delta Y$, $\delta Z$ is the error of $EC_{50}$ values X, Y and Z and $\delta R$ is the calculated error of the predicted effect (R)). Hashed bars reflect the experimentally measured effect of each combination of mutations tested. (**C**) The effect of recreating the most plausible evolutionary trajectory for 20_39 with (solid bars) or without (hashed bars) the addition of H215N. In all figures an unpaired t-test was used to determine whether there was a significant difference in chloramphenicol detoxification activity between two groups. (***, $p \leq 0.001$; **, $p \leq 0.01$; *, $p \leq 0.05$). Data presented in all figures represent the average of at least four biological repeats ± 1 S.D.

The online version of this article includes the following source data for figure 3:

**Source data 1.** Source data for *Figure 3*.

neither of which conferred a significant improvement in chloramphenicol detoxification when introduced to NfsA individually (*Figure 3A*) or together (*Figure 3B*). When each was introduced into an R225V or R225D background, S41Y yielded a significant increase in chloramphenicol detoxification, but T219Y did not (*Figure 3B*). However, the combination of S41Y and T219Y together with R225V or R225D gave a further significant improvement (*Figure 3B*). Numerous examples of sign epistasis can also readily be observed in the full network diagram (e.g. the blue circles indicate a negative impact for certain combinations of substitutions; *Figure 2D–E*, *Figure 2—figure supplement 1*, *Figure 2—figure supplement 2*). For example, H215N (present in variant 20_39) is detrimental to chloramphenicol detoxification activity when substituted into the R225D or R225D/S41Y backgrounds, and somewhat neutral in combination with R225D/S41Y/T291Y, but significantly enhances activity in combination with R225D/S41Y/T291Y/F227H (*Figure 3C*). Overall, our data suggest that complex epistatic interactions render >99% of the 5040 hypothetical evolutionary pathways (that might be traversed from wild-type NfsA to either 36_37 or 20_39) broadly inaccessible to iterative mutagenesis strategies.

## Improved chloramphenicol detoxification is underpinned by loss of activity with 1,4-benzoquinone

The hypothetical evolutionary trajectories depicted in *Figure 2F* highlight particularly pertinent intermediate combinations of mutations. We considered that interrogating the intermediate variants might shed light on the mechanistic basis of improved chloramphenicol detoxification. In particular, we wanted to determine how activity with a presumed native substrate like 1,4-benzoquinone was affected during the hypothetical evolutionary progression towards improved chloramphenicol detoxification. For this, the enzyme intermediates identified in the most probable stepwise evolutionary trajectory (*Figure 2F*) were purified as His-tagged proteins and in vitro kinetics assays attempted with both chloramphenicol and 1,4-benzoquinone (*Supplementary file 1b*, *Figure 2D–E*, *Figure 4—figure supplement 1*). From these data, it was evident that the first substitution of both hypothetical trajectories (i.e. the elimination of R225) was sufficient to abolish nearly all 1,4-benzoquinone activity. A structure of NfsA with 1,4-benzoquinone in the active site was reported from the Hyde group that proposes the quinone is held in place through interactions between one of its carbonyl oxygens and the backbone of S41, while the other carbonyl oxygen interacts with the guanidinium of R225 and/or the amine of Q67 (*Day, 2013*). Our data are consistent with R225 playing a key role in binding this substrate.

Although both the R225V and R225D variants exhibited just-detectable levels of activity at a high concentration of 1,4-benzoquinone (100 μM; *Figure 4A and B*, *Supplementary file 1b*), no activity was discernible at lower concentrations, precluding measurement of kinetic parameters. The level of 1,4-benzoquinone activity remained unmeasurably low with all subsequent substitutions (*Figure 4A and B*). The sustained loss of 1,4-benzoquinone activity throughout each trajectory is consistent with elimination of quinone competition promoting more effective chloramphenicol reduction. This is reinforced by examination of the chloramphenicol detoxification activities of the complete set of hypothetical evolutionary intermediates (*Figure 2D–E*, *Figure 2—figure supplement 1*, *Figure 2—figure supplement 2*); the variants that retained R225 were on average no better than wild-type NfsA at defending host cells against chloramphenicol (mean fold improvement of 1.0 ± 0.5), while cells expressing variants that contained the substitution R225V or R225D were on average able to tolerate 4.0 ± 2.4 fold higher chloramphenicol concentrations than those expressing wild-type *nfsA* (*Supplementary file 1c*).

The substitution S41Y that came next in both trajectories yielded a profound improvement in chloramphenicol $K_M$, but also diminished $k_{cat}$ substantially (*Figure 4B and C* and *Figure 4F and G*). We observed the same S41Y NfsA substitution in our previous PR-104A study, and concluded that this most likely enables planar stabilisation and stacking of nitroaromatic substrates between the isoalloxazine rings of flavin mononucleotide (FMN) and the introduced tyrosine (*Copp et al., 2017*). It is likely that a similar phenomenon explains the improved affinity for chloramphenicol observed here, with the decrease in catalytic turnover also arising as a consequence of enhanced stabilisation of the Michaelis complex. The subsequent substitutions in each trajectory then act to 'tune' the system, exerting only minor effects on $k_{cat}$, but overall yielding incremental improvements in chloramphenicol $K_M$ that largely mirror the improved chloramphenicol detoxification observed in vivo (*Figure 4C and E* and *Figure 4D and F*). SDS-PAGE analysis confirmed that the expression levels

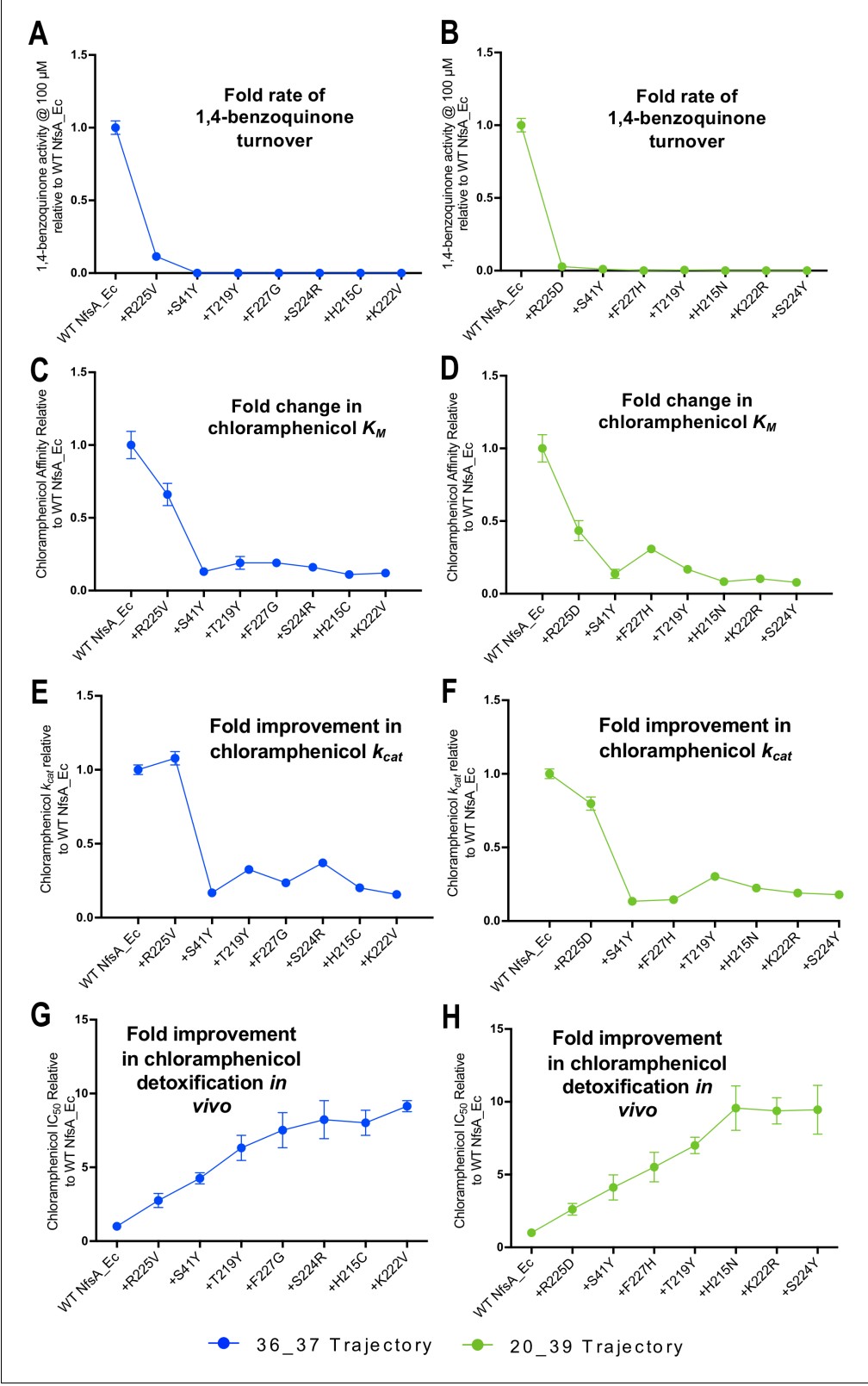

**Figure 4.** Activity analysis with 1,4-benzoquinone (A, B) and chloramphenicol (C–H) during the hypothetical evolutionary progression of 36_37 (left, blue) and 20_39 (right, green). (A, B) Fold rate of turnover of 1,4-benzoquinone (starting concentration 100 µM, with 250 µM NADPH co-substrate) relative to wild-type NfsA for each intermediate variant of 36_37 (A) and 20_39 (B). (C, D) Fold change in chloramphenicol $K_M$ relative to wild-type NfsA for each intermediate of 36_37 (C) and 20_39 (D). (E, F) Fold increase in chloramphenicol $k_{cat}$ relative to wild-type NfsA for each intermediate of 36_37

*Figure 4 continued on next page*

*Figure 4 continued*

(E) and 20_39 (F). (G, H) Fold improvement in chloramphenicol detoxification (EC$_{50}$) conferred to *E. coli* 7NT host cells by each variant relative to wild-type NfsA, reproduced for convenience from *Figure 2F*. Full Michaelis-Menten kinetic parameters are shown in *Supplementary file 1b*, and the raw Michaelis-Menten curves are presented in *Figure 4—figure supplement 1*. All in vitro data presented is the average of three technical repeats ± 1 S.D. and all in vivo data presented in the average of at least four biological repeats ± 1 S.D.

The online version of this article includes the following source data and figure supplement(s) for figure 4:

**Source data 1.** Source data for *Figure 4*.

**Figure supplement 1.** Michaelis-Menten plots of chloramphenicol reduction by NfsA enzyme intermediates derived from the most plausible hypothetical evolutionary trajectories for variants 36_37 and 20_39.

**Figure supplement 2.** Relative enzyme expression levels for key intermediates from the most plausible hypothetical evolutionary trajectories for (A) 36_37 and (B) 20_39.

---

were consistent for each intermediate variant throughout the hypothetical evolutionary progression, eliminating this as a variable exerting substantial influence on the relative activity levels in vivo (*Figure 4—figure supplement 2*).

## Directed evolution of NfsA via epPCR yields exclusively R225-substituted variants

The above data suggest that selection of NfsA variants for enhanced chloramphenicol detoxification within *E. coli* was underpinned by substitution of R225, eliminating competition with endogenous quinones, after which the improved chloramphenicol affinity conferred by additional substitutions became relevant. However, while our targeted mutagenesis strategy allowed us to comprehensively explore a defined region of sequence space, we could not rule out that other evolutionary solutions might be possible. For example, we considered that an alternative spectrum of mutations might improve chloramphenicol detoxification without eliminating quinone reduction. To investigate this, we used epPCR to introduce mutations across the entire *nfsA* coding sequence, then transformed *E. coli* 7NT cells and subjected them to selection on LB agar amended with 10 µM chloramphenicol.

Growth inhibition assays were performed for 60 randomly chosen clones at a low (7.5 µM), medium (15 µM) or high (30 µM) concentration of chloramphenicol. These assays confirmed that all selected variants were able to detoxify chloramphenicol, with two appearing comparable to wild-type *nfsA* and the remainder providing more substantial host protection (*Supplementary file 1d*). When each gene variant was sequenced, it was revealed that there were 50 unique sequences, each of which encoded a protein that was substituted at the R225 position (*Supplementary file 1e*). In contrast, sequencing of 20 randomly-chosen clones grown without chloramphenicol selection identified no R225 substitutions, confirming that the epPCR library was not profoundly biased towards mutation of codon 225 (*Supplementary file 1f*). We were surprised to note (*Supplementary file 1e*) that codon 225 was mutated exclusively at the first nucleotide position, giving substitutions R225C (C→T, 47 times), R225S (C→A, seven times) or R225G (C→G, six times), but not at the second nucleotide position (which would have given rise to R225L/P/H). Of the latter three possibilities, R225L and R225H were both encoded by the degenerate NDT triplet used in our original combinatorial mutagenesis (*Figure 1B*), but R225L was only recovered once in our top 30 variants and R225H not at all (*Supplementary file 1e*). It may be that as individual substitutions, none of R225L/P/H confer sufficient improvement in chloramphenicol detoxification to have been selected in this experiment. Irrespective, that 100% of selected variants were mutated at codon 225 provides compelling evidence that eliminating quinone competition is a key first step towards evolving improved chloramphenicol detoxification, and that substitution of R225 is the most attainable way to achieve this.

## Impact of improved affinity for chloramphenicol on alternate substrates

We were interested to discover how selection for improved activity with chloramphenicol had impacted unselected promiscuous activities of NfsA. We therefore used EC$_{50}$ growth assays to assess the sensitivities of *E. coli* 7NT strains individually expressing either NfsA, variants 36_37 or 20_39, or the hypothetical evolutionary intermediates thereof, to five structurally diverse nitroaromatic prodrugs (*Figure 5*). We anticipated that the loss of competitive inhibition by endogenous quinones might have generically enhanced activity with each of these prodrugs, resulting in heightened host cell toxicity. However, for four of the five prodrugs, host sensitivity was largely unchanged

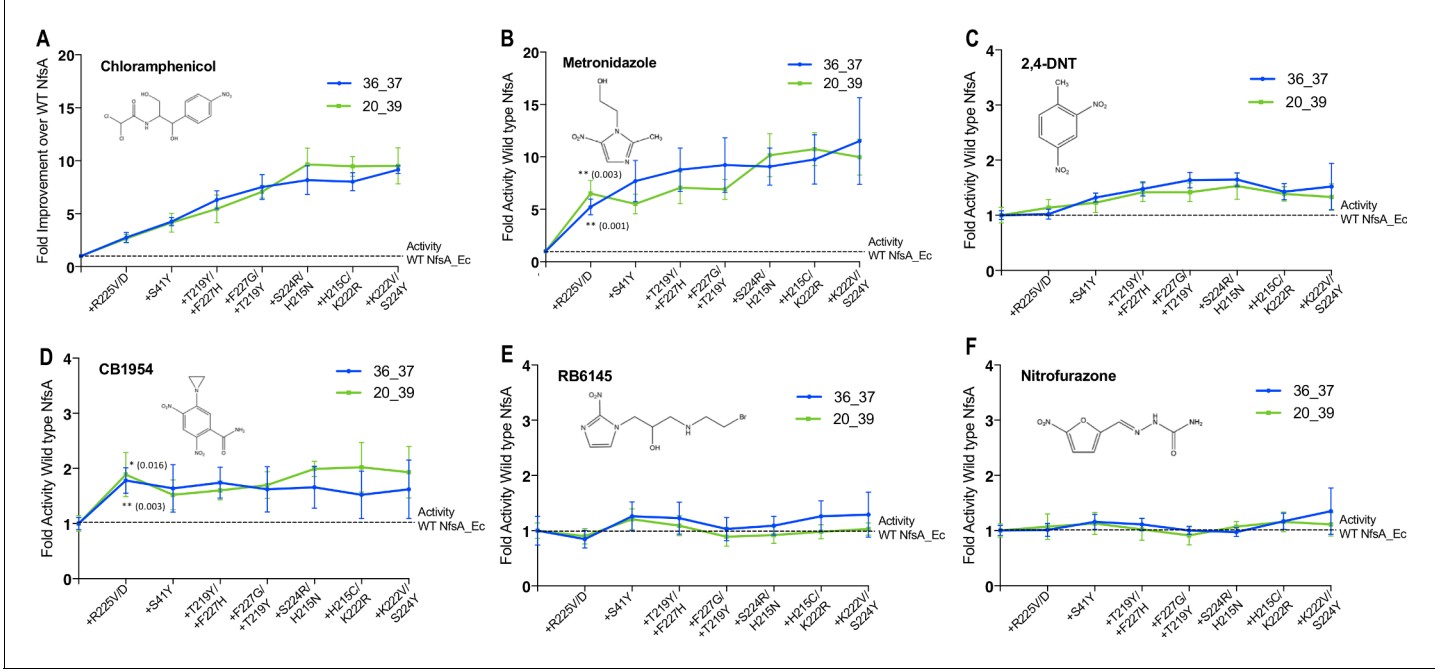

**Figure 5.** Activity analysis with nitroaromatic prodrugs during the hypothetical evolutionary progression of 36_37 (blue) and 20_39 (green). *E. coli* 7NT cells expressing each of the hypothetical intermediate variants of 36_37 and 20_39 were tested in $EC_{50}$ growth assays for (**A**) resistance to chloramphenicol; and (**B–F**) sensitivity to metronidazole, 2,4-DNT, CB1954, RB6145 and nitrofurazone, respectively. Data is presented as the fold improvement relative to wild-type NfsA, from the average of four biological repeats ± 1 S.D. Chloramphenicol (**A**) and metronidazole (**B**) data are plotted on a different scale due to the large fold improvements in activity. Where substitution of R225 caused a significant improvement in prodrug activation (Student's t-test), the p-value is noted in the figure panel (above the trendline for variant 20_39, and below for variant 36_37).

The online version of this article includes the following source data and figure supplement(s) for figure 5:

**Source data 1.** Source data for *Figure 5*.
**Figure supplement 1.** Growth of 36_37 and 20_39 on selective or counter-selective media.

($EC_{50}$ within a range of 0.8 to 2-fold that of the NfsA-expressing strain) when expressing any of the variants (*Figure 5C–F*). The exception was metronidazole, for which all variants exhibited similar gains in activity to chloramphenicol, despite the two compounds sharing little structural similarity (*Figure 5A–B*). Moreover, the introduction of R225V or R225D substitutions into NfsA (which largely eliminate 1,4-benzoquinone activity; *Figure 4A–B*) did not improve reduction of all prodrugs, but only significantly enhanced activity with metronidazole and CB1954 (Student's t-test; *Figure 5B,D*). We therefore concluded that our selection for enhanced chloramphenicol detoxification was not driven exclusively by loss of the competing quinone activity, as this would have tended to also enhance activity with other alternate substrates.

## Applications of 36_37 and 20_39 as dual selectable/counter- selectable marker genes

Whereas reduction of chloramphenicol is a detoxifying activity, reduction of metronidazole yields a toxic product. The serendipitous gains in metronidazole sensitivity that paralleled improved chloramphenicol detoxification inspired us to investigate whether these opposing activities might have useful molecular biology applications, by offering dual selectable and counter-selectable functionalities in a single gene. Counter-selectable markers, such as the *sacB* levansucrase gene from *Bacillus subtilis*, have multiple applications including the forced elimination of plasmids, and resolution of merodiploid constructs during allele exchange (*Stibitz, 1994*). However, they must typically be partnered with a selectable marker on the same DNA construct, to enable positive selection for the construct before its subsequent elimination. This occupies additional space, which is undesirable for size-restricted constructs, and means there is potential for the two genes to become separated by recombination events, leading to false positive or false negative outcomes.

Because metronidazole is cheap, widely-available, and has no measurable bystander effect in *E. coli* (i.e. unlike many other nitroaromatic prodrugs its toxic metabolites are confined solely to the activating cell [*Chan-Hyams et al., 2018*]), we considered it ideally suited for counter-selection applications. We therefore tested the abilities of chloramphenicol to maintain, or metronidazole to force elimination of, plasmids bearing either 36_37 or 20_39 in *E. coli* 7NT. Cells were cultured for one hour in the absence of any selective compound, then plated on solid media amended with either 5 µM chloramphenicol or 10 µM metronidazole. The resulting colonies were then tested for retention or loss of the plasmid, respectively, with the expected outcome being realised in 100% of cases (94/94 colonies tested; *Figure 5—figure supplement 1*). This suggests that our selected variants might indeed have useful applications as dual selectable/counter-selectable marker genes.

## Discussion

By exploiting a powerful selection for antibiotic resistance, we were able to implement simultaneous mass-mutagenesis on an unprecedented scale, to amplify a promiscuous functionality. We acknowledge that this approach of focusing exclusively on eight key active-site residues means the reconstructed NfsA 'engine' is unlikely to be an optimal fit within the pre-existing chassis, and that further gains in activity would undoubtedly result by selecting residue substitutions in the second shell, or beyond. Nevertheless, we reasoned that our approach would allow us to gain comprehensive insight into key catalytic changes driving improved chloramphenicol detoxification, without being subject to the stochastic vagaries of epPCR, or the well-established phenomenon that it can only access a limited and unbalanced repertoire of residues (on average, only 5.7 of the 19 alternative amino acids per codon position, with a bias towards similar residues [*Hermes et al., 1990*]). We also initially considered that this approach might allow us to leap to a fitness peak that iterative random mutagenesis strategies would be unable to scale. However, when we examined every hypothetical evolutionary intermediate, we discovered this was not the case; although stepwise evolution would have been greatly constrained in the progression from wild-type NfsA to either of our top two variants, there were plausible trajectories to achieve these outcomes. Notably, the critical first step in any of these trajectories was the substitution of R225, leading to near-total abrogation of the native quinone reductase activity, which was never restored and appears incompatible with the selected activity. The importance of eliminating R225 was emphasised by epPCR mutagenesis of *nfsA*, when 50/50 unique variants selected for improved chloramphenicol detoxification were found to have lost R225, while 20/20 unselected variants retained it (*Supplementary files 1e,f*). The strong trade-off between quinone and chloramphenicol reduction is a very different scenario to the predominantly weak trade-offs observed in previous laboratory evolution studies (e.g. those reviewed by *Kaltenbach et al., 2016*). Even the more recent work of *Ben-David et al., 2020* who encountered an abrupt activity trade-off when they evolved the calcium-dependent lactonase mammalian paraoxonase-1 into an efficient organophosphate hydrolase, found that the native functionality could subsequently be restored and was not incompatible with the evolved one (*Ben-David et al., 2013*; *Ben-David et al., 2020*).

By choosing to select for a novel enzymatic activity within the native cellular environment, we deliberately set out to explore the additional complexities of metabolic interference, which have potential to play dominant roles in shaping natural evolutionary outcomes. *Copley, 2020* recently described an equation that succinctly summarises how the rate of a promiscuous reaction in the presence of a native substrate might be improved by (1) increasing the concentration of the enzyme; (2) increasing the ratio of promiscuous to native substrate; and/or (3) altering the active site to diminish substrate competition, by enhancing binding or turnover of the promiscuous substrate, or decreasing binding of the native substrate. In a landmark 2015 study she and co-workers experimentally demonstrated the importance of diminished substrate competition, focusing on a single key Glu to Ala substitution that enabled several orthologs of ProA (L-gamma-glutamyl phosphate reductase, a key enzyme in proline synthesis) to replace *E. coli* ArgC (an N-acetyl glutamyl phosphate reductase required for arginine synthesis) (*Khanal et al., 2015*). Where measurable, all of the substituted variants showed decreased affinity (increased $K_M$) for the native substrate; and in all but one case there was substantial improvement in $k_{cat}/K_M$ for the promiscuous substrate as well (*Khanal et al., 2015*). These findings are similar to our observation that elimination of quinone reductase activity from NfsA via substitution of R225 provided a platform for successive

improvements in chloramphenicol affinity to amplify host cell resistance. Together, these examples support the proposal of *Kaltenbach et al., 2016* that during natural evolution of a promiscuous activity there is likely to be active selection against the original function, as well as our own supposition that most previous laboratory evolution studies have evaded this phenomenon by focusing on heterologous enzymes and/or exogenously applied substrates. Moreover, our observations that the unselected promiscuous activities of NfsA (reduction of a structurally diverse panel of prodrugs) were mostly unaffected is consistent with their central thesis, that positive selection alone does not lead to specialisation. The emerging picture is that evolution in the natural intracellular milieu involves both selection for the new function, and selection against the old.

An interesting difference between our scenario and that of the Copley team is that NfsA is less essential to the fitness of its host cell than ProA (e.g. deletion of *nfsA* does not impair *E. coli* growth even under oxidative stress from heavy metal challenge [*Ackerley et al., 2004*]). This means that when a new stress is encountered and a promiscuous function becomes essential, as we have modelled here, the enzyme can potentially evolve without necessitating gene duplication to preserve the original function. An apparent 'freedom to operate' is manifest in the vast diversity of primary functionalities observed in the superfamily of nitroreductases that NfsA belongs to (which spans activities as divergent as quinone reduction, flavin reduction to power bioluminescence, flavin fragmentation, dehalogenation and dehydrogenation [*Akiva et al., 2017*]). Although this contrasts with the prevailing Innovation–Amplification–Divergence (IAD) model for natural enzyme evolution (*Bergthorsson et al., 2007*), it may not be an exceptional scenario – rather, as previously argued by *Newton et al., 2015* it is likely that only a minority of enzymes in a cell are under active selection for improved catalytic activity at any time, and redundancy in metabolic networks means that there is latent evolutionary potential that can be immediately tapped to adapt to stress without the requirement of rare and costly gene duplication events. That a single mutation may suffice to rapidly amplify a desirable promiscuous activity simply by eliminating native substrate competition confers substantial 'robustness' at a cellular level, even if it means that individual enzymes may not be as robust as previously considered.

## Materials and methods

### Chemicals

Chloramphenicol, metronidazole, 2,4-dinitrotoluene and nitrofurazone were purchased from Sigma-Aldrich. CB1954 was purchased from MedKoo Biosciences. RB6145 was synthesised in-house at the Ferrier Institute, Victoria University of Wellington.

### Simultaneous site-directed mutagenesis library construction and selection

To randomise the eight targeted residues of NfsA_Ec (S41, L43, H215, T219, K222, S224, R225, and F227) we designed a degenerate gene construct with NDT codons (specifying Arg, Asn, Asp, Cys, Gly, His, Ile, Leu, Phe, Ser, Tyr, and Val) at all positions other than 219 (NHT codon, encoding Ala, Asn, Asp, His, Ile, Leu, Phe, Pro, Ser, Thr, Tyr, and Val) and 222 (VNG codon, encoding Ala, Gln, Glu, Gly, Leu, Lys, Met, Pro, Thr, Val, and two Arg codons). Initially a synthetic gene library was ordered from Lab Genius pre-cloned into plasmid pUCX (*Prosser et al., 2013*), however, this only yielded 15% of the 252 million unique variants in our final collection. The remaining 85% were generated ourselves by ordering the same sequence as a gene fragment library from GenScript and ligating it into pUCX at the *NdeI* and *SalI* restriction sites. The combined libraries were used to transform *E. coli* 7NT, a derivative of strain W3110 bearing gene deletions of seven endogenous nitroreductases (*nfsA, nfsB, azoR, nemA, yieF, ycaK* and *mdaB*) and the *tolC* efflux pump (*Copp et al., 2014a*). Electrocompetent *E. coli* 7NT cells were generated as per *Sambrook and Russell, 2001*, and the transformation efficiency was enhanced using a yeast tRNA protocol modified from *Zhu and Dean, 1999*. Library selection was conducted on selective solid media containing LB agar supplemented with 100 µg.mL$^{-1}$ ampicillin and either 45 or 47.5 µM chloramphenicol. Appropriate dilutions of the pooled library stock were spread over plates and incubated at 37°C for 40 hr. Dilutions of the library were also spread over non-selective solid media (LB agar supplemented with 100 µg.mL$^{-1}$ ampicillin) to estimate the number of transformants included in each selection. Enzyme intermediates of NfsA_Ec

36_37 and 20_39 were ordered as synthetic gene fragments from Twist Biosciences and subsequently ligated into the *NdeI* and *SalI* restriction sites of the vectors pUCX (for $EC_{50}$ analysis) or pET28(a)$^+$ (for purification of His$_6$-tagged proteins).

## Growth assays

For growth inhibition assays, a 96-well microtitre plate with wells containing 200 µL LB medium supplemented with 0.2% glucose (w/v) and 100 µg.mL$^{-1}$ ampicillin was inoculated with *E. coli* 7NT nitroreductase strains and incubated at 30°C with shaking at 200 rpm for 16 hr. A 15 µL sample of overnight culture was used to inoculate 200 µL of induction media (LB supplemented with 100 µg.mL$^{-1}$ ampicillin, 0.2% (w/v) glucose and 50 µM IPTG) in each well of a fresh microtitre plate, which was then incubated at 30°C, 200 rpm for 2.5 hr. Aliquots of 30 µL apiece from these cultures were used to inoculate four wells of a 384-well plate (two wells containing 30 µL induction media and two wells containing 30 µL induction media supplemented with 2 × the desired chloramphenicol concentration). The cultures were incubated at 30°C, 200 rpm for 4 hr. Cell turbidity was monitored by optical density at 600 nm prior to drug challenge and 4 hr post- challenge. The percentage growth inhibition was determined by calculating the relative increase in $OD_{600}$ for challenged versus control wells.

For $EC_{50}$ growth assays, 100 µL of overnight cultures as above were used to inoculate 2 mL of induction media and incubated at 30°C, 200 rpm for 2.5 hr. A 30 µL sample of each culture was added to wells of a 384-well plate containing 30 µL of induction media supplemented with 2 × the final prodrug concentration. Each culture was exposed to 7–15 drug concentrations representing a 1.5-fold dilution series of drug and one unchallenged (induction media only) control. The cultures were incubated at 30°C, 200 rpm for 4 hr. Cell turbidity was monitored by optical density at 600 nm prior to drug challenge and 4 hr post-challenge. The $EC_{50}$ value of technical replicates was calculated using a dose-response inhibition four-parameter variable slope equation in GraphPad Prism 8.0. The $EC_{50}$ values of biological replicates were averaged to provide a final $EC_{50}$ value.

## Evolutionary trajectory analysis

Full details of how the evolutionary trajectory analysis was conducted are available at https://github.com/MarkCalcott/Analyse_epistatic_interactions/tree/master/Create_mutation_network (*Calcott, 2020*; copy archived at swh:1:rev:62624a66324e230bf273b17591a537c691fa316f).

## Protein purification and steady-state kinetics

Recombinant nitroreductases were cloned into the His$_6$-tagged expression vector pET28(a)$^+$, expressed in BL21 and purified as His$_6$-tagged proteins. Enzyme reactions were carried out in 60 µL reactions in 96-well plates with a 4.5 mm pathlength. All reactions were performed in 10 mM Tris HCl buffer pH 7.0, 250 µM NADPH, an appropriate dilution of chloramphenicol or 1,4-benzoquinone substrate dissolved in DMSO (0–4000 µM chloramphenicol and 100 µM 1,4-benzoquinone), made up to volume with ddH$_2$O. Reactions were initiated with the added of 6 µL of enzyme (8 µM or an appropriate concentration) and the linear decrease in absorbance was monitored at 340 nm measuring the rate of NADPH depletion as an indirect measured of substrate reduction. As neither chloramphenicol nor 1,4-benzoquinone interfere with the absorbance at 340 nm, the extinction coefficient of NADPH at 340 nm was used (chloramphenicol = 12,400 M$^{-1}$cm$^{-1}$ and *p*-benzoquinone = 6,220 M$^{-1}$cm$^{-1}$, as two molecules of NADPH are required to reduce chloramphenicol to the hydroxylamine form, while only one is required for the reduction of *p*-benzoquinone to the quinol). Technical replicates were plotted using Graphpad Prism 8.0 software and non-linear regression analysis and Michaelis-Menten curve fitting was performed.

## SDS-PAGE analysis of *nfsA* variant expression levels

*E. coli* 7NT pUCX::*nfsA* variant strains were used to inoculate 200 µL LB media supplemented with 0.2% (w/v) glucose and Amp. Cultures were incubated overnight at 30°C, 200 rpm. The next day, 100 µL of the overnight culture was used to inoculate 2 mL of LB induction medium (LB supplemented with 0.2% (w/v) glucose, Amp and 50 µM IPTG). Day cultures were grown at 30°C, 200 rpm for 6.5 hr, after which the cultures were pelleted by centrifugation at 2500 × *g* for 5 min. The supernatant was decanted and the cell pellets resuspended in ~100 µL of LB medium, after which the

$OD_{600}$ of a 1:100 dilution was measured. Cell cultures were normalised by dilution with additional LB medium so that a 1 in 100 dilution would give an $OD_{600}$ reading of 0.1. A 12 µL sample of each culture was mixed with 5 × SDS loading buffer, heated at 95°C for 5 min and subjected to SDS-PAGE analysis on a 15% acrylamide gel.

### Directed evolution of *nfsA* via epPCR

Error-prone PCR of NfsA_Ec was performed using a Gene Morph II kit (Agilent) as described by *Copp et al., 2014b*. An error rate of 3.25 mutations per amplicon (calculated from *Supplementary file 1f*) was achieved by using 35 amplification cycles from 1.5 ng of purified PCR product as template per 25 µL reaction (prepared using primers pUCX_fwd; GACATCATAACGG TTCTG and NfsA_rev; GGGTCGACTTAGCGCGTCGCCCAACCCTG). Amplicon size and quality were assessed by agarose gel electrophoresis, then amplicons were digested with NdeI and SalI and ligated into similarly-digested pUCX plasmid. The resulting ligation was used to transform the screening strain *E. coli* 7NT, generating a library of approximately $3 \times 10^6$ variants. Selection of improved variants from this library was carried out as described for the multi-site directed mutagenesis library above, but on LB agar plates supplemented with 10 µM chloramphenicol as well as 5 µM IPTG and 100 µg.mL$^{-1}$ ampicillin.

### Evaluating the selection/counter-selection potential of evolved *nfsA* variants

A single colony of an *E. coli* 7NT cells expressing *nfsA*_Ec 36_37 or 20_39 was used to inoculate a 3 mL overnight culture of LB supplemented with 100 µg.mL$^{-1}$ ampicillin. The next day, 100 µL of each overnight culture was used to inoculate 10 mL fresh LB medium in a 125 mL baffled conical flask. The culture was grown at 37°C, 200 rpm for 1 hr then the $OD_{600}$ of the flask was determined. An appropriate dilution of each culture was plated on agar plates containing either LB-only, or LB amended with 10 µM metronidazole or 5 µM chloramphenicol. At 10 µM metronidazole, cells expressing NfsA_Ec 36_37 or 20_39 could not grow but cells bearing no plasmid could, while the reverse scenario applied with 5 µM chloramphenicol. Plates were incubated at 37°C for 16 hr (LB-only or LB + metronidazole) or 40 hr (LB + chloramphenicol). To confirm the presence/absence of the plasmid bearing 36_37 or 20_39, 47 colonies from each condition were streaked on LB agar plates supplemented with 100 µg.mL$^{-1}$ ampicillin and incubated at 37°C for 16 hr, with growth indicating presence of the plasmid and no growth indicating absence of the plasmid. The same 47 colonies, alongside a negative control were further tested in a PCR screen with *nfsA*_Ec forward and reverse specific primers (*Prosser et al., 2013*). A band of approximately 720 bp indicated presence of the plasmid, while no band indicated absence of the plasmid.

### Statistical analysis

Unless otherwise stated, data are given as the mean ± standard deviation. The software programme GraphPad Prism 8.0 was used for all statistical analyses. Differences between measured $EC_{50}$ values of enzyme variants were determined by an unpaired Student's t-test.

A p-value of ≤0.05 was considered statistically significant with ***p-value ≤0.001, **p-value ≤0.01 and *p-value ≤0.05.

## Acknowledgements

We thank Professor Dan Tawfik for insightful suggestions on shaping the research and Associate Professor Nobu Tokuriki for his comments on an early draft of the manuscript. This research was funded by The Royal Society of New Zealand Marsden Fund (contracts 19-VUW-076 and 15-VUW-037 to DFA and WMP, including a PhD scholarship for KRH). MHR received additional support from a Victoria University of Wellington (VUW) Doctoral Scholarship, RFL a VUW Masters Scholarship, and MJC from a research grant awarded by the Cancer Society of New Zealand (grant 18.05 to MJC and DFA).

## Additional information

### Funding

| Funder | Grant reference number | Author |
|---|---|---|
| Royal Society of New Zealand | 15-VUW-037 | Wayne M Patrick<br>David F Ackerley |
| Cancer Society of New Zealand | 18.05 | Mark J Calcott<br>David F Ackerley |
| Royal Society of New Zealand | 19-VUW-076 | Wayne M Patrick<br>David F Ackerley |
| Victoria University of Welling-ton | Doctoral Scholarship | Michelle H Rich |
| Faculty of Science, Victoria University of Wellington | VUW Masters Scholarship | Rory F Little |

The funders had no role in study design, data collection and interpretation, or the decision to submit the work for publication.

### Author contributions

Kelsi R Hall, Conceptualization, Data curation, Formal analysis, Validation, Investigation, Visualization, Methodology, Writing - original draft, Writing - review and editing; Katherine J Robins, Conceptualization, Data curation, Formal analysis, Investigation, Methodology; Elsie M Williams, Rory F Little, Validation, Methodology; Michelle H Rich, Validation, Investigation, Methodology; Mark J Calcott, Data curation, Software; Janine N Copp, Validation, Methodology, Writing - review and editing; Ralf Schwörer, Resources; Gary B Evans, Resources, Supervision; Wayne M Patrick, Conceptualization, Resources, Supervision, Funding acquisition, Writing - review and editing; David F Ackerley, Conceptualization, Resources, Formal analysis, Supervision, Funding acquisition, Writing - original draft, Project administration, Writing - review and editing

### Author ORCIDs

Katherine J Robins  https://orcid.org/0000-0001-5049-4246
Michelle H Rich  https://orcid.org/0000-0003-4876-4029
Mark J Calcott  https://orcid.org/0000-0002-7736-8095
Janine N Copp  https://orcid.org/0000-0001-6690-0480
Ralf Schwörer  https://orcid.org/0000-0002-9352-6559
Wayne M Patrick  https://orcid.org/0000-0002-2718-8053
David F Ackerley  https://orcid.org/0000-0002-6188-9902

### Decision letter and Author response

Decision letter https://doi.org/10.7554/eLife.59081.sa1
Author response https://doi.org/10.7554/eLife.59081.sa2

## Additional files

### Supplementary files

• Source data 1. Supplementary CAM growth inhibi.

• Supplementary file 1. Supplementary files 1a-1f. (a) Summary of substitutions present in the top 30 chloramphenicol-detoxifying variants generated by multi-site saturation mutagenesis. (b) Kinetic parameters of chloramphenicol reduction and turnover rate of 1,4-benzoquinone for intermediates from the most plausible hypothetical evolutionary trajectories for (A) 36_37 and (B) 20_39. Apparent $K_M$ and $k_{cat}$ were calculated using Graphpad 8.0. Kinetic parameters could not be accurately determined for 1,4-benzoquinone, therefore the catalytic rate of 1,4-benzoquinone reduction was measured at a single high concentration of 1,4-benzoquinone (100 µM) with reactions initiated by addition of 250 µM NADPH. All reactions were measured in triplicate and errors are ±1 S.D. In the

left-most column, the terminology '+' refers to an enzyme variant that has the same amino acid sequence as the variant in the row above, plus the one additional substitution indicated. For example, '+R225V' describes a variant sharing an identical primary sequence to NfsA, with the additional substitution R225V. *Apparent $k_{cat}$ and $K_M$ as determined at 250 µM NADPH. **Measured rates following addition of 250 µM NADPH. ***N.D.=not detectable (change in $OD_{340}$ <0.1 s$^{-1}$). (c) Average fold improvement for all NfsA_Ec variants that either retained R225 or contained a R225V/D substitution. To calculate the average fold improvement of variants retaining R225, the fold improvement relative to wild-type NfsA_Ec of all 64 variants retaining R225 was averaged. To calculate the average fold improvement of variants with the R225V or R225D substitutions, the fold improvement relative to wild-type NfsA_Ec of all 64 variants containing either R225V (in 36_37 intermediates) or R225D (in 20_39 intermediates) was averaged. (d) Relative levels of chloramphenicol growth inhibition experienced by *E. coli* 7NT strains expressing the 50 unique *nfsA* variants generated by epPCR. Following transformation of *E. coli* 7NT cells with the epPCR library, plating on LB amended with 10 µM chloramphenicol, and random selection of 60 colonies, 50 unique variants (numbered from ep_1 to ep_50; ep for 'error-prone PCR') were identified by Sanger sequencing of the pUCX inserts. Fresh day cultures of each unique strain were incubated at 30°C, 200 rpm for 4 hr post-challenge with either a low (7.5 µM), medium (15 µM) or high (30 µM) concentration of chloramphenicol, and percentage growth inhibition was determined by calculating the relative increase in $OD_{600}$ for challenged cultures relative to unchallenged replicates. (e) Summary of all encoded amino acid and nucleotide substitutions identified in the 50 unique *nfsA* variants obtained from the epPCR library following chloramphenicol selection. (f) Summary of all encoded amino acid and nucleotide substitutions identified in 20 randomly-chosen *nfsA* variants obtained from the epPCR library in the absence of chloramphenicol selection.

- Transparent reporting form

### Data availability
All data generated or analysed during this study are included in the manuscript and supporting files.

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
