## [Decision Letter]

**Acceptance summary:**

The potential for endogenous native substrates to interfere with evolution of the new function has so far been mostly a matter of speculation and your paper now shows conclusively that in the quinone reductase NfsA, a promiscuous chloramphenicol-detoxifying nitroreductase activity can only be amplified if a key first mutation eliminates the native activity of NfsA, after which additional substitutions incrementally improve the affinity for chloramphenicol as a substrate. This example substantiates a model for enzyme evolution, in which a single mutation can open the path to the rapid amplification of a desirable promiscuous activity simply by eliminating native substrate competition.

**Decision letter after peer review:**

Thank you for submitting your article "A giant leap in sequence space reveals the intracellular complexities of evolving a new function" for consideration by *eLife*. Your article has been reviewed by three peer reviewers, and the evaluation has been overseen by Andrei Lupas as the Reviewing Editor and Patricia Wittkopp as the Senior Editor. The reviewers have opted to remain anonymous.

The reviewers have discussed the reviews with one another and the Reviewing Editor has drafted this decision to help you prepare a revised submission.

Summary:

This manuscript describes a laboratory evolution experiment designed to explore effects that may shape evolutionary trajectories in a native host environment. The model system is *E. coli* nitro/quinone reductase NfsA, a promiscuous FMN-dependent oxidoreductase that reduces toxic compounds and has the basal ability to reduce the antibiotic chloramphenicol. This function was used to select for improved detoxification by mass-mutagenizing eight active-site residues and isolating variants with up to 10-fold higher tolerance against chloramphenicol. The five best variant proteins were purified and characterized, showing that their kcat/Km was only marginally improved, with worse kcat but improved Km, indicating that the improvements in detoxification were driven by enhanced substrate affinity. For the top two variants, all possible evolutionary trajectories were recreated and their EC_50_'s tested to determine the most likely possible step-wise paths from NfsA to the final variants. The authors found that iterative evolutionary strategies could have generated similar variants, but that there were only few accessible pathways, indicating epistatic effects. The analysis also showed that for both variants, elimination of arginine at position 225 in the first step enabled further improvements to take hold and played a role in the loss of wildtype 1,4-benzoquinone activity. The sensitivity to four out of five tested prodrugs was however unchanged. Turnover of the fifth prodrug, namely reduction of metronidazole, which yields a toxic product, was on the other hand increased in the evolved variants, and could be used as a counter-selectable marker. This was briefly tested showing the potential of such an application.

Essential revisions:

This study presents a wealth of data, and is well reasoned, carefully executed and clearly laid out. However, although it states that its aim was to study the evolution of a promiscuous function within the native host environment and thus under metabolic interference of the native substrate, this was not the approach taken. Instead, a fitness peak for the promiscuous function was identified through mass mutagenesis at eight positions followed by selection, and then two potential evolutionary paths leading from the wild type to this peak were inferred based on an analysis of all possible mutant combinations at the mutagenized positions. The authors need to make clear throughout the paper that the variants able to detoxify chloramphenicol were not evolved and did not arise against metabolic interference of the native substrate. This is an important point as the considerable potential of endogenous metabolites to shape evolutionary outcomes (Abstract) is purely inferred from the observation that the first mutation in both reconstructed evolutionary paths appears to have been a mutation at R225, which led to a substantial drop in the turnover rate of the endogenous substrate. From this the authors conclude (very prominently throughout the paper) that the evolution of a new activity is only possible after loss of activity against the original substrate. From the data presented, it is however not clear to what extent this conclusion is supported.

1) According to the data in Figure 4 and Supplementary file 1B, mutation of R225 alone is accompanied by a ~2-fold increase in kcat/Km for chloramphenicol. This seems to be sufficient to explain the ~2-fold increase in EC_50_ for chloramphenicol without invoking loss of quinone reductase activity. The control experiment in Figure 5, showing that substitution of R225 has no effect on most promiscuous activities of NfsA, also seems to indicate that the loss of native activity is not required for the evolution of chloramphenicol resistance. It would be important to determine the kinetic parameters of 1,4-benzoquinone reduction for NfsA and the purified R225V and R225D mutants in order to establish the loss of quinone reductase activity in the postulated first step of the evolutionary path. It would also be useful to study the effect of 1,4-benzoquinone competition on the chloramphenicol reductase activity of the mutants, at least the first ones along the proposed path, in order to show that they rapidly become insensitive to the native substrate.

2) After the initial screening of the transformed library of NfsA variants, 0.05% of gene variants are reported to be more effective in chloramphenicol detoxification than the wild type. In the next steps, this number is reduced to the top 30 variants, as characterized by their improvement of chloramphenicol EC_50_ values (Figure 1D). However, it is not clear from the presentation whether these observations were controlled for the expression levels of the different NfsA mutants. Protein variants are often expressed at different levels in vivo, which can have a significant effect on the activity measured. Figure 1D was used for selection of the "best" variants for the rest of the study and to support this choice and the conclusions of the manuscript, relative enzyme expression levels should be reported (and if significantly different, should be corrected for). Such expression levels are reported later on for the 36_37 and 20_39 variants, but are missing at this early stage.

3) While mutation of R225 appeared to be required for improved chloramphenicol detoxification in this study, the authors only considered the effects of substitutions at eight positions. This is probably the main weakness of the combinatorial mutagenesis approach used here. It seems plausible that substitutions at other positions could also increase chloramphenicol tolerance, possibly opening a path without loss of quinone reductase activity. If the authors were able to perform one round of error-prone PCR on NfsA with selection for improved chloramphenicol resistance and obtain mainly variants with substitution of R225, this would substantially strengthen their claim that evolution of increased chloramphenicol resistance can only occur through loss of quinone reductase activity.

4) Even with additional experimental support for the main conclusion of the article, it seems fundamentally problematic to extrapolate from two instances to a general principle of evolution. The authors should tone done the claims that improved chloramphenicol detox activity is ONLY possible after elimination the native activity and instead comment on the two characterized mutant pathways as examples of this phenomenon, within the limitations of the experimental setup.

---

## [Author Response]

Essential revisions:This study presents a wealth of data, and is well reasoned, carefully executed and clearly laid out. However, although it states that its aim was to study the evolution of a promiscuous function within the native host environment and thus under metabolic interference of the native substrate, this was not the approach taken. Instead, a fitness peak for the promiscuous function was identified through mass mutagenesis at eight positions followed by selection, and then two potential evolutionary paths leading from the wild type to this peak were inferred based on an analysis of all possible mutant combinations at the mutagenized positions. The authors need to make clear throughout the paper that the variants able to detoxify chloramphenicol were not evolved and did not arise against metabolic interference of the native substrate. This is an important point as the considerable potential of endogenous metabolites to shape evolutionary outcomes (Abstract) is purely inferred from the observation that the first mutation in both reconstructed evolutionary paths appears to have been a mutation at R225, which led to a substantial drop in the turnover rate of the endogenous substrate. From this the authors conclude (very prominently throughout the paper) that the evolution of a new activity is only possible after loss of activity against the original substrate. From the data presented, it is however not clear to what extent this conclusion is supported.

Thank you for raising this point. We recognise the distinction between our mass mutagenesis strategy and stepwise evolution, and have edited our manuscript to eliminate any unjustified claims regarding evolution of our variants (most commonly, we added the word “hypothetical” in front of “evolutionary trajectory” , and substituted the word “evolved” with “selected” where the former was not justified). We do however disagree with the statement that our variants “did not arise against metabolic interference of the native substrate”. A key aspect of our work that is different from many previous directed evolution studies is that we selected our improved variants within their native host environment, where they were very specifically exposed to their native substrates. We consider that this clearly shaped the outcomes.

With credit to the excellent suggestion from the reviewers that we perform epPCR on the native *nfsA* gene to see whether substitution of R225 predominates, we note that we now have much stronger evidence to support our conclusions around the need to eliminate the native activity to enable improved chloramphenicol detoxification (see response to reviewers’ point 3 below).

1) According to the data in Figure 4 and Supplementary file 1B, mutation of R225 alone is accompanied by a ~2-fold increase in kcat/Km for chloramphenicol. This seems to be sufficient to explain the ~2-fold increase in EC_50_ for chloramphenicol without invoking loss of quinone reductase activity. The control experiment in Figure 5, showing that substitution of R225 has no effect on most promiscuous activities of NfsA, also seems to indicate that the loss of native activity is not required for the evolution of chloramphenicol resistance. It would be important to determine the kinetic parameters of 1,4-benzoquinone reduction for NfsA and the purified R225V and R225D mutants in order to establish the loss of quinone reductase activity in the postulated first step of the evolutionary path. It would also be useful to study the effect of 1,4-benzoquinone competition on the chloramphenicol reductase activity of the mutants, at least the first ones along the proposed path, in order to show that they rapidly become insensitive to the native substrate.

There are two suggestions here, which will be addressed in turn.

i) We had in fact reported our efforts to obtain 1,4-benzoquinone reduction kinetics for the purified R225V and R225D variants in addition to wild type NfsA, however these data were evidently buried a bit too deeply in the legend to Supplementary file 1B for them to be readily accessed by a reader. In brief, it was not possible for us to derive meaningful 1,4-benzoquinone kinetic parameters for the two variants (both R225V and R225D exhibit a just-detectable level of 1,4-benzoquinone activity, as was indicated in the “Rate with 100 µM *p*-benzoquinone” column of Supplementary file 1B and in Figures 4A and B; but the activity was essentially undetectable at lower concentrations). This is the basis for our conclusion that quinone reductase activity has indeed been lost in the postulated first step of the evolutionary path. To more overtly justify this conclusion to the reader, we have now referred to these data explicitly in the main text, as detailed in our response to the next point.

ii) We agree that it would be ideal to study the effect of 1,4-benzoquinone competition on the chloramphenicol reductase activity of the mutants, but unfortunately both chloramphenicol and 1,4-benzoquinone reduction assays rely on monitoring NADPH depletion at 340 nm, so it is not possible to deconvolute the relative turnover of each substrate by spectrophotometric assays in a mixed solution (in our previous study, we were able to independently monitor PR-104A reduction at 400 nm). However, it is important to note that in our previous study the top selected variant was still partially active with 1,4-benzoquinone in vitro, making it plausible that substrate competition could occur. Here we show that 1,4-benzoquinone activity had effectively been eliminated from our R225-substituted variants, and we therefore consider it reasonable to presume that 1,4-benzoquinone will no longer exhibit substrate competition with chloramphenicol. We have now clarified these points in the text, with the last two sentences of the relevant paragraph now reading:

“Here, we were unable to perform the same in vitro competition assays, as both 1,4-benzoquinone and chloramphenicol reduction are monitored by following NADPH depletion at 340 nm. […] In contrast, for wild type NfsA, 1,4-benzoquinone reduction (*k_cat_*/*K_M_* = 5.8 × 10^6^; Valiauga et al., 2017) is nearly 1000-fold more efficient than chloramphenicol reduction (Table 1). Our data are therefore consistent with in vivo chloramphenicol reduction having been amplified for the selected variants by the elimination of competitive quinone inhibition.”

We also note that the additional epPCR experiments (see point 3 below) strongly reinforce that evolving improved chloramphenicol detoxification necessitates elimination of R225, which is consistent with 1,4-benzoquinone outcompeting chloramphenicol as a substrate for wild type NfsA.

2) After the initial screening of the transformed library of NfsA variants, 0.05% of gene variants are reported to be more effective in chloramphenicol detoxification than the wild type. In the next steps, this number is reduced to the top 30 variants, as characterized by their improvement of chloramphenicol EC_50_ values (Figure 1D). However, it is not clear from the presentation whether these observations were controlled for the expression levels of the different NfsA mutants. Protein variants are often expressed at different levels in vivo, which can have a significant effect on the activity measured. Figure 1D was used for selection of the "best" variants for the rest of the study and to support this choice and the conclusions of the manuscript, relative enzyme expression levels should be reported (and if significantly different, should be corrected for). Such expression levels are reported later on for the 36_37 and 20_39 variants, but are missing at this early stage.

We agree that this is an important variable that merits consideration. We have now performed SDS-PAGE analysis for the *E. coli* strains expressing each of the top 30 variants alongside wild type NfsA controls, and can confirm that there is no substantial variation in expression between these variants and wild type NfsA. We have provided these data as a supplementary figure (new Figure 1—figure supplement 1), and have added the following sentence to the main text: “SDS-PAGE analysis revealed that there was no substantial variation in expression between these variants and native *nfsA*, and hence the enhanced chloramphenicol detoxification was not a consequence of elevated expression levels Figure 1—figure supplement 1).”

3) While mutation of R225 appeared to be required for improved chloramphenicol detoxification in this study, the authors only considered the effects of substitutions at eight positions. This is probably the main weakness of the combinatorial mutagenesis approach used here. It seems plausible that substitutions at other positions could also increase chloramphenicol tolerance, possibly opening a path without loss of quinone reductase activity. If the authors were able to perform one round of error-prone PCR on NfsA with selection for improved chloramphenicol resistance and obtain mainly variants with substitution of R225, this would substantially strengthen their claim that evolution of increased chloramphenicol resistance can only occur through loss of quinone reductase activity.

This was a really fantastic suggestion, and we kicked ourselves for not having previously thought to run this experiment. Our previous conclusions regarding the need to eliminate the native quinone reductase activity have now received a compelling corroboration by our discovery that 100% of unique selected variants (50/50) after one round of epPCR on native *nfsA* were carrying an R225 substitution. We have added an additional section of text entitled “Directed evolution of NfsA via epPCR yields exclusively R225-substituted variants” and two additional supplementary tables (Supplementary file 1E and F) and a new supplementary figure to describe these findings; and have reworked our Abstract and Discussion slightly to recognise them.

4) Even with additional experimental support for the main conclusion of the article, it seems fundamentally problematic to extrapolate from two instances to a general principle of evolution. The authors should tone done the claims that improved chloramphenicol detox activity is ONLY possible after elimination the native activity and instead comment on the two characterized mutant pathways as examples of this phenomenon, within the limitations of the experimental setup.

Thanks to the excellent suggestion above (major point 3), we now have far stronger evidence to support our primary conclusion. Nevertheless, we have tried to tone down our previous claims, as detailed using track changes throughout the manuscript.